

# Multi-decadal retreat of marine-terminating outlet glaciers in northwest and central-west Greenland

Taryn E. Black[1,2], Ian Joughin[2]

[1]Department of Earth and Space Sciences, University of Washington, Seattle, 98195, United States
5   [2]Polar Science Center, Applied Physics Laboratory, University of Washington, Seattle, 98105, United States

*Correspondence to*: Taryn E. Black (teblack@uw.edu)

**Abstract.** The retreat and acceleration of marine-terminating outlet glaciers in Greenland over the past two decades has been widely attributed to climate change. Here we present a comprehensive annual record of glacier terminus positions in northwest and central-west Greenland and compare it against local and regional climatology to assess the regional sensitivity 10   of glacier termini to different climatic factors. This record is derived from optical and radar satellite imagery and spans 87 marine-terminating outlet glaciers from 1972 through 2021. We find that in this region, most glaciers have retreated over the observation period, and widespread regional retreat accelerated around 1996. The acceleration of glacier retreat coincides with the timing of sharp shifts in ocean surface temperatures, duration of sea-ice season, ice-sheet surface mass balance, and meltwater and runoff production. Our findings suggest that a variety of processes – such as ocean-interface melting, mélange 15   presence and rigidity, and hydrofracture-induced calving – contribute to, but do not conclusively dominate, the observed regional retreat.

## 1 Introduction

The Greenland Ice Sheet has lost significant mass over the last few decades (Enderlin et al., 2014; Shepherd et al., 2020) as many of its glaciers have retreated (Hill et al., 2017; Howat and Eddy, 2011; Moon and Joughin, 2008; Murray et al., 2015; 20   King et al., 2020), and ice flow has accelerated (Moon et al., 2012; Rignot and Kanagaratnam, 2006; Joughin et al., 2010). Recent Greenland ice loss has contributed to rates at times approaching 1 mm a$^{-1}$ of global sea-level rise (Shepherd et al., 2020), with the contribution from northwest and central-west Greenland combined representing nearly half of the cumulative contribution from Greenland to sea-level rise since 1972 (Mouginot et al., 2019). Although surface mass balance has dominated Greenland's mass loss in the past two decades, in northwest and central-west Greenland over half of the mass loss 25   is currently due to ice discharge (Mouginot et al., 2019), which has accelerated since 2000 in this region (King et al., 2020). Changes in ice discharge are often related to changes to glacier terminus positions, with terminus retreat into deeper water driving acceleration and up-stream thinning (Howat et al., 2008; Joughin et al., 2008b). Because of the relationship between terminus position and calving rate, initial perturbations that increase calving can trigger further glacier retreat. While glacier



retreat and acceleration are generally linked to changes at the terminus, it remains unclear which processes are most

responsible for controlling perturbations to calving rates and subsequent terminus retreat (Straneo et al., 2013).

The recent acceleration and retreat of Greenland outlet glaciers has been attributed to warmer ocean temperatures (Holland et al., 2008; Howat et al., 2008; Morlighem et al., 2016; Wood et al., 2021; Fahrner et al., 2021; Rignot et al., 2012; Slater et al., 2019), to changes in sea ice and mélange characteristics (Amundson et al., 2010; Carr et al., 2013; Cassotto et al., 2015; Foga et al., 2014; Joughin et al., 2008a; Moon et al., 2015), and to increased melting and crevassing associated with warming

air temperatures (Benn et al., 2007; Nick et al., 2013; van der Veen, 1998). In turn, responses to these forcings are modulated by factors associated with individual glaciers such as bed topography and fjord width (Carr et al., 2015; Schild and Hamilton, 2013; Catania et al., 2018; Felikson et al., 2021). Because of the interaction of terminus position with glacier geometry, detailed records of terminus position changes for many glaciers are necessary to identify the importance of different forcing mechanisms on decadal-scale outlet glacier changes across a large area.

Most previous studies of Greenland outlet glacier terminus positions have been temporally or spatially limited. Some studies cover the entire ice sheet for over a decade, but map termini only decadally or in non-consecutive years (Howat and Eddy, 2011; Moon and Joughin, 2008). Other studies map termini more frequently, but only for a small sector of the ice sheet (Bjørk et al., 2012; McFadden et al., 2011; Moon et al., 2015) or for a few specific glaciers (Joughin et al., 2008b; Holland et al., 2016; Larsen et al., 2016; Motyka et al., 2017; Schild and Hamilton, 2013). Murray *et al.* (2015) mapped terminus

positions at high spatial and temporal resolutions, but only for a single decade. More recent studies have attempted to fill these observational gaps with ice sheet-wide analyses of annual terminus positions spanning multiple decades (Wood et al., 2021; Fahrner et al., 2021; King et al., 2020), although they come to differing conclusions about the drivers of observed terminus retreat.

In this paper, we analyze glacier change at high spatiotemporal resolution by constructing a multi-decadal (49 years) record

of annual terminus positions for 87 marine-terminating outlet glaciers in northwest and central-west Greenland. This record allows us to identify the behavior of individual glaciers as well as regional trends in the magnitude and timing of glacier retreat. We compare this regional behavior with climate data – sea surface and subsurface temperatures, sea-ice concentration, and ice-sheet surface mass balance, precipitation, melt, and runoff – to assess the relative influence of different forcing mechanisms on multi-decadal and annual terminus retreat in this sector of Greenland. It is important to note

that we do not account for the effect of geometric factors such as bed topography on modulating the retreat of individual glaciers, as we instead focus on terminus retreat in correspondence with regional climate trends.

## 2 Data and Methods

We used 455 synthetic aperture radar (SAR) and optical satellite images to trace annual terminus positions for 87 marine-terminating outlet glaciers in northwest and central-west Greenland (68.9° N to 78.2° N) from 1972 through 2021.



## 2.1 Satellite Images

We digitized terminus positions from SAR images acquired by the European Copernicus Program's Sentinel-1A/B and the Canadian Space Agency's Radarsat-1 satellites for 13 of the 49 years in our record. These satellites use radar, which allows them to image the surface regardless of clouds or darkness. For the most recent seven years (2015–2021), we traced terminus positions in mosaics of Copernicus Sentinel-1A/B images (Joughin et al., 2016a). These mosaics are typically created from images acquired in early February of their respective years and have 25 or 50-m product-dependent resolution. Radarsat-1 mosaics were used to trace terminus positions for six winters (2000–2001, 2005–2009, 2012–2013) (Moon and Joughin, 2008; Joughin et al., 2015). These mosaics are formed from images collected from October through March and have nominal resolutions of 20 m.

For the remaining 36 years in our record, we used imagery from all Landsat missions to map terminus positions. We also used Landsat imagery to map individual glacier termini where they were missing from or indiscernible in the SAR imagery. We prescreened images in the USGS Global Visualization Viewer (GloVis) to confirm that glaciers of interest were not obscured by clouds and that the images were georeferenced well. Due to winter darkness, we could not select images from the same time of year as the SAR mosaics; instead, we chose images as close to winter as possible, with a preference for spring over fall to capture a more winter-like state to reduce the effects of seasonal variation. As a result, the majority of the Landsat images we used were collected in spring (March–May). Because of the difficulty in finding sunlit, cloud-free imagery, however, we had to use data from other periods, so all months except January and December are represented. For some glaciers there are several years with missing data. The image resolution ranges from 15 m (Landsat 8 panchromatic band) to 60 m (Landsat 1-5 Multispectral Scanner).

## 2.2 Terminus Positions

Using ArcGIS, we manually traced annual terminus positions from the radar and Landsat imagery. This dataset builds on a preexisting dataset covering six winters between 2000 and 2013 (Moon and Joughin, 2008; Joughin et al., 2015). The study area ranges from Saqqarliup Sermia (68.9° N, 50.3° W; ~35 km southwest of Jakobshavn Isbræ) to Bamse Gletsjer (78.2° N, 72.7° W) (Fig. 1). Individual glacier names and coordinates are detailed in Supplementary Table 1. We selected all marine-terminating outlet glaciers that are at least ~1.5 km wide at the terminus and are flowing at least ~1000 m a$^{-1}$. We traced each glacier's terminus position once per hydrological year (September 1 through August 31 (Ettema et al., 2009)) in every year for which suitable imagery was available. We used winter or near-winter imagery whenever possible as indicated by Figure 2.

Errors in terminus position may arise from both the imagery used and the digitization process. The primary sources of errors introduced by the image data are the uncertainties in position after orthorectification and georeferencing. To reduce such errors, candidate images were compared with control images and discarded if they were noticeably offset or distorted. Manual tracing introduces errors, which are exacerbated by lack of image resolution and image artifacts (such as shadows or



an indistinct transition from glacier to mélange) (Joughin et al., 2015). If a terminus position was ambiguous in one image, it was flagged during tracing, and compared with close-in-time images from the same or other satellite platforms when possible. Digitization errors are typically comparable to image resolution; for example, 25 m error for a Landsat-7 image

with 30 m resolution (Moon et al., 2015)



**Figure 1:** Map of study area, showing individual glacier locations (white) and points where ocean data were acquired (orange, numbered). Glacier labels are (a) every tenth glacier, numbered, and (b) specific glaciers named in the paper, acronymized as



follows: **Jakobshavn Isbræ (JI), Alianaatsup Sermia (AS), Sermeq Avannarleq (SA), Store Gletsjer (ST), Sermilik Isbræ (SI), Kangilleq (KA), Kangerlussuup Sermia (KS), Rink Isbræ (RI), Upernavik Isstrøm (UI), Naajarsuit Sermiat (NS), unnamed glacier #26 (26), Alison Glacier (AL), unnamed glacier #36 (36), Kjer Gletsjer (KJ), Kong Oscar Gletsjer (KO), unnamed glacier #55 (55), unnamed glacier #66 (66), Tracy Gletsjer (TR), Verhoeff Gletsjer (VH), and Morris Jesup Gletsjer (MJ). Basemaps are (a) bed topography from BedMachine Greenland V3 (Morlighem et al., 2017b, a) and (b) ice-sheet velocity from MEaSUREs GIMP (Joughin et al., 2016b, 2018a).**

In addition to the errors associated with digitization of the images, there is additional uncertainty introduced by sampling seasonal variability at different times of year. For the terminus positions mapped from Landsat imagery, it was not possible to get sunlit and cloud-free images over each glacier at the same time every year (Fig. 2). This timing variation complicates the year-to-year comparison at a glacier, which might include seasonal variability in terminus position. Such seasonal variability could produce some short-term deviations in our data; for example, Moon *et al.* (2015) found a mean annual

terminus range of 610 m for a subset of our study glaciers. However, the data collection season is largely consistent across each of our glaciers (Fig. 2) and because these seasonal errors are not independent, they tend to cancel out over longer periods. Since we mostly focus on decadal scale trends, issues of seasonal sampling should not greatly affect our results.

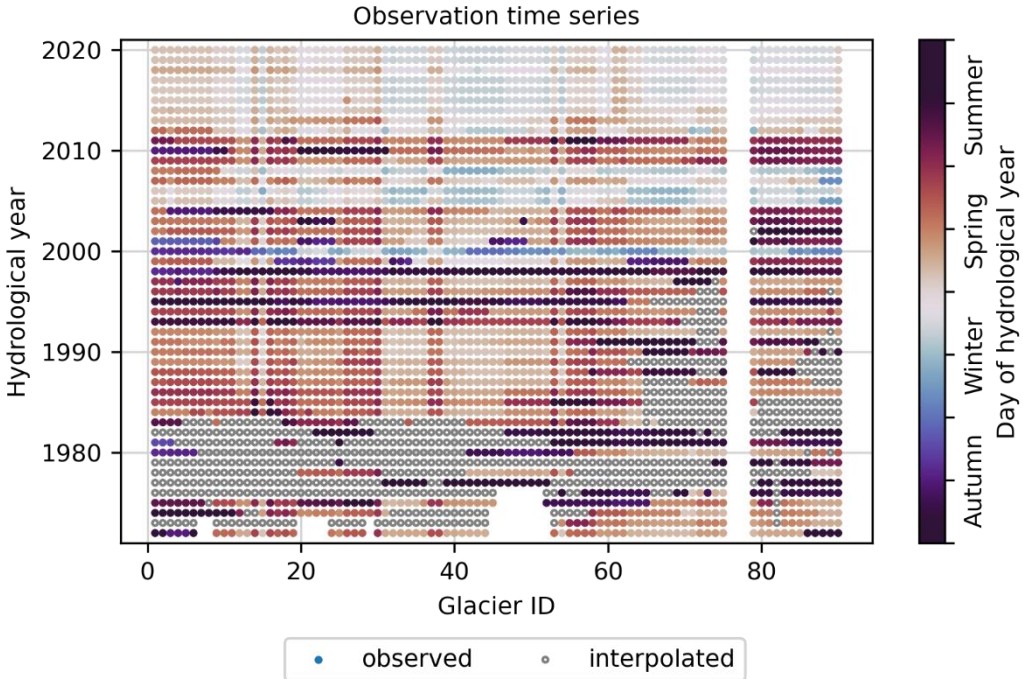

**Figure 2: Years in which a terminus position was observed (filled circles) for each glacier, colored by the season of the observation.**
**If a terminus position was not observed, an estimated position was interpolated (open circles) from prior and subsequent observations to use in annual analyses.**

## 2.3 Glacier Change Measurements

We calculated glacier area change over time using the box method (Moon and Joughin, 2008). Each glacier has a static, open-ended reference box (polygon) that approximately delineates the main region of ice flow. The box sides are roughly





parallel to ice flow, and the 'back' of the box is perpendicular to ice flow at an arbitrary location up-glacier of the extent of
maximum observed terminus retreat (see example in Fig. 3). Where a terminus trace intersects the open end of the box, the
polygon is closed, and the area of that polygon represents the glacier's reference area at that point in time. Repeating this
process for each terminus trace for a glacier forms a time series of reference areas, from which we determine the annual area
change. While the areas are arbitrarily determined by the box size, the area differences between successive terminus traces

represent the annual gain or loss of area. By focusing on the area change between measurements rather than the absolute area
of each measurement, we can ignore the arbitrariness of how the boxes are drawn. There is a small error associated with
these area change measurements because the boxes do not completely conform to the glacier sides.

As length change can provide a more intuitive measure of retreat than area change, we determine the nominal length change
by scaling the area change by the average width of the box to get an approximate length change. This measurement should

be interpreted as a proxy for length change rather than an exact measurement. Compared to the centerline method of
measuring length change, this method is less sensitive to uncertainties in the centerline position at the terminus.

In years when no observations were made, we linearly interpolated between the prior and subsequent observations to
estimate glacier length and area during the missing years (Fig. 2). For glaciers with missing observations in the beginning of
the record, we did not interpolate prior to the first observation in the record. The largest temporal gap interpolated is nine

years (1975–1976 through 1983–1984), at Sermeq Avannarleq (#8). Most temporal gaps are in the 1970s and early 1980s,
with additional gaps into the 1990s for some high-latitude glaciers, and there are no temporal gaps after 2002.

Because it is difficult to compare changes in area between glaciers of different sizes, we also determined the percent area
change over time for each glacier. For each individual glacier area time series, we normalized the minimum observed glacier
area to 0 and the maximum observed glacier area to 1, and linearly scaled the other measured areas between those set points.

This method normalizes every glacier's area change to the same 0–1 scale, allowing straightforward comparison of size
changes between different glaciers. Because the equivalent length is simply a scaled area, the results are identical for area
and length.

To pinpoint the timing of changes in glacier area and length, we performed a break-point analysis on the time series for each
glacier. We fit each time series with a piecewise-linear function with two segments (Jekel and Venter, 2019); the break point

between the two segments corresponds to a year in the time series that is the best fit for approximating the time series as a
two-segment piecewise linear function. Although this method always finds one break point in the time series, even if the
number of break points is greater or there is no break point at all, we adopted it in order to identify the singular important
year for each glacier when a change in behavior may have occurred.

**2.4 Climate Data**

We acquired several ice-sheet and oceanographic datasets in order to compare our glacier terminus position changes with
climatic factors. Ice-sheet surface mass balance, snowfall, rainfall, meltwater production, and runoff came from the Modèle
Atmosphérique Régional (MAR), Version 3.11, on a 6km x 6km grid over the period 1979–2020.

Sea-surface temperatures came from the NASA-JPL Estimating the Circulation and Climate of the Ocean (ECCO) consortium ocean circulation model, Version 5 (Forget et al., 2015; Zhang et al., 2018), on a 1/3° x 1/3° grid over the period 155 1992–2017, and the merged Hadley-OI sea-surface temperature and sea-ice concentration dataset (Hurrell et al., 2008; Shea et al., 2020), on a 1° x 1° grid over the period 1972–2020.

We also obtained subsurface temperatures from ECCO for most of our study area. In the Disko Bay area we used field observations from the ICES Dataset on Ocean Hydrography (ICES, 2014), collected over the period 1977–2016. For both data sets, subsurface temperatures were sampled at approximately 250 m depth.

Sea-ice concentration came from the merged Hadley-OI dataset, as well as the NOAA/NSIDC Climate Data Record of Passive Microwave Sea Ice Concentration, Version 3 (Peng et al., 2013; Meier et al., 2017), on a 25km x 25km grid over the period 1978–2019. For the sea-ice concentration, we calculated the annual duration of the period when sea-ice concentration exceeds 15%, as well as the seasonal mean sea-ice concentration.

For each variable and dataset, we considered the annual and decadal mean values. Due to lack of reliable data in narrow 165 fjords, we used offshore observations as a proxy for fjord conditions. For ocean and sea-ice data, we extracted the annual and decadal mean values at eight points offshore (Fig. 1), selected to be representative of clusters of glaciers: Disko Bay, Uummannaq Fjord, Upernavik Icefjord and north, south Melville Bay/Wilcox Head, central Melville Bay, north Melville Bay/Cape York, Wolstenholme Bay/Thule, and Inglefield Fjord (coordinates for each point are detailed in Supplementary Table 2). For the ice-sheet variables, we considered the annual mean near the front of each glacier, as well as the population 170 annual and decadal means.

## 3 Results

We produced a comprehensive multi-decadal dataset of terminus positions for glaciers in our study area, and collected climate data both near the termini of these glaciers and in the ocean offshore of clusters of these glaciers.

### 3.1 Terminus Positions

We created a dataset of 3606 annual terminus positions for our 87 selected glaciers from 1972–1973 through 2020–2021 (see example in Fig. 3). The median number of annual observations per glacier is 41, and nearly all glaciers were observed in 38 to 46 of the 49 years examined. Only three glaciers have fewer (33–34) observations; these glaciers are located just south of Thule Air Force Base and have limited imagery available in the 1980s and early 1990s. After interpolating area changes between glacier observations, the first year with either an observation or an interpolation available for each glacier is 1977–180 1978. Therefore, our glacier analyses start in this year except where noted.



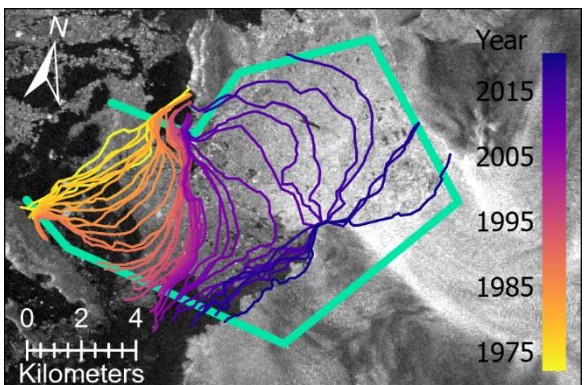

**Figure 3: Illustration of terminus traces and box method for Kjer Gletsjer (#42). Each trace intersects the glacier's box (green) to form a closed polygon. The areas of sequential polygons are differenced to create a time series of glacier area change. The basemap is a Sentinel-1 SAR mosaic from February 4–9, 2020.**

### 3.1.1 Terminus Behavior

The majority of the glaciers in our study area retreated between 1977 and 2021. The cumulative area loss of all of these glaciers is 1067 km$^2$, equivalent to a cumulative retreat of 287 km. The individual area and length changes are plotted in Fig. 4 with glaciers with the greatest change broken out separately. We identify these dominant glaciers as those with a net change falling more than 2 standard deviations beyond the population mean, which yields four glaciers that dominate the area change: Jakobshavn Isbræ (#3; -130.9 km$^2$), Alison Glacier (#35; -59.4 km$^2$), Kjer Gletsjer (#42; -81.5 km$^2$), and Tracy Gletsjer (#81; -58.7 km$^2$). These glaciers together are responsible for 31.0% of the total area loss. Five glaciers dominate the length change: Jakobshavn Isbræ (-16.9 km), Alison Glacier (-14.7 km), Kjer Gletsjer (-14.7 km), Tracy Gletsjer (-14.0 km), and one unnamed glacier (#36; -11.1 km). These glaciers are cumulatively responsible for 24.9% of the total retreat. For the remaining glaciers, the mean area change is -8.9 km$^2$ and the mean length change is -2.6 km. Net area and length changes for individual glaciers are detailed in Supplementary Table 3.

Fifteen glaciers were stable in that the net changes in area were within 2 standard deviations of their respective observed interannual variability, and no glaciers advanced significantly over the observation period. The fifteen stable glaciers are: Alianaatsup Sermia (#7), Sermeq Avannarleq (#8), Store Gletsjer (#9), Sermilik (#11), Kangilleq (#12), Kangerlussuup Sermia (#16), Rink Isbræ (#17), Upernavik Isstrøm (#21), Naajarsuit Sermiat (#25), Kong Oscar Gletsjer (#51), Verhoeff Gletsjer (#86), Morris Jesup Gletsjer (#87), and unnamed glaciers #26, #55, and #66.





**Figure 4: Cumulative (a) area change and (b) length change for large glaciers that dominate the total observed change. Glacier changes in each plot are labeled using the acronym scheme in Figure 1. The colormap follows the order of the glaciers (from south to north), so glaciers with similar colors are spatially closer together. For all other glaciers, cumulative (c) area change and (d) length change are provided but not distinguished by glacier. The final net (e) area change and (f) length change are reported as histograms.**





### 3.1.2 Timing of Change

Because a small number of glaciers tend to dominate the overall trends in glacier changes, we scaled each glacier's length and area to a 0–1 scale as described above in order to consistently compare the relative timing of each glacier's behavior

(Fig. 5a). The resulting data are noisy, so to better identify overall trends we computed the mean and standard deviation of the scaled glacier changes (blue overlay in Fig. 5a). Of the observed mean size change (area or equivalent length), ~4% occurs each decade before 1996, and ~31% occurs each decade after 1996. The population of glaciers notably re-advanced in 2017 and 2018, but began to retreat again in 2019.

We performed a break-point analysis (see Sect. 2.3) for each glacier's area time series (Fig. 5b). The step-change in retreat

rates around 1996 for the normalized time series is consistent with the time series break points for individual glaciers. The most common break point years were 1996–1997 and 1997–1998, with additional smaller peaks in break points in the mid-1990s and mid-2000s. The break point years for each glacier are detailed in Supplementary Table 3.

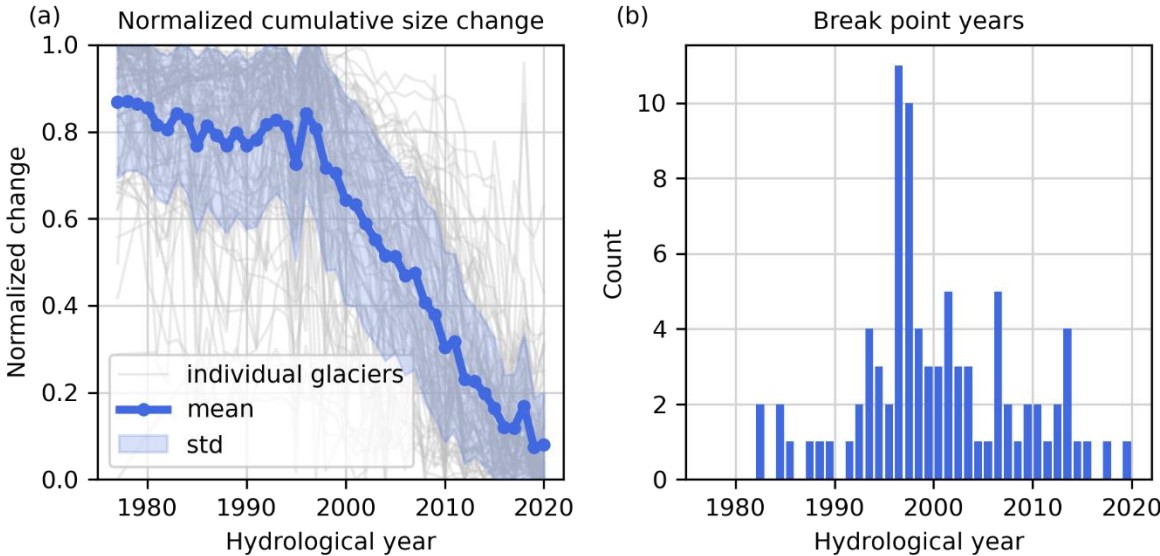

**Figure 5: Timing of change. (a) Glacier cumulative area change is normalized such that for each glacier the greatest observed**
**extent is 1 and the smallest observed extent is 0, so that glaciers of all sizes are placed on a common scale. Each glacier's cumulative change is then approximated as a two-segment piecewise linear function, and we compute (b) a histogram of the breakpoint year between the two segments.**

### 3.2 Climate Data

Figure 6 summarizes the data from the MAR climate model. While there is considerable interannual variability, the data

indicate a net decrease in surface mass balance (Fig. 6a) and a net increase in both meltwater production (Fig. 6b) and runoff (Fig. 6c), with corresponding increases in the annual number of meltwater production days (Fig. 6d) and runoff days (Fig. 6e), since the late 1970s. The mean surface mass balance was steady in the 1980s and 1990s, but dropped by 0.48 m a$^{-1}$ between the 1990s and the 2000s, primarily due to substantial increases in meltwater production (+0.42 m a$^{-1}$) and runoff



(+0.48 m a$^{-1}$). By contrast, there were only small changes to the mean snowfall (-0.01 m a$^{-1}$; Fig. 6f) and the mean rainfall

(+0.02 m a$^{-1}$; Fig. 6g) from the 1980s to the 2010s. For the 2010s, while the annual surface mass balance anomaly was positive in 2013, 2017, and 2018, it reached its lowest values for the full record in 2019 and 2020. These extremes are reflected in coincident meltwater production and runoff anomalies. For the decade as a whole, the decadal mean surface mass balance anomaly is slightly more negative than that of the 2000s, and the decadal mean meltwater production and runoff continued to increase.

Like the mean meltwater production and runoff, the mean number of melt days (Fig. 6d) and runoff days (Fig. 6e) increased between the 1990s and the 2000s by 7 days and 12 days, respectively. However, instead of continuing to increase in the 2010s, the number of melt and runoff days decreased, suggesting that the net increase in meltwater production and runoff was due to more intense melt events rather than a greater number of melt events.



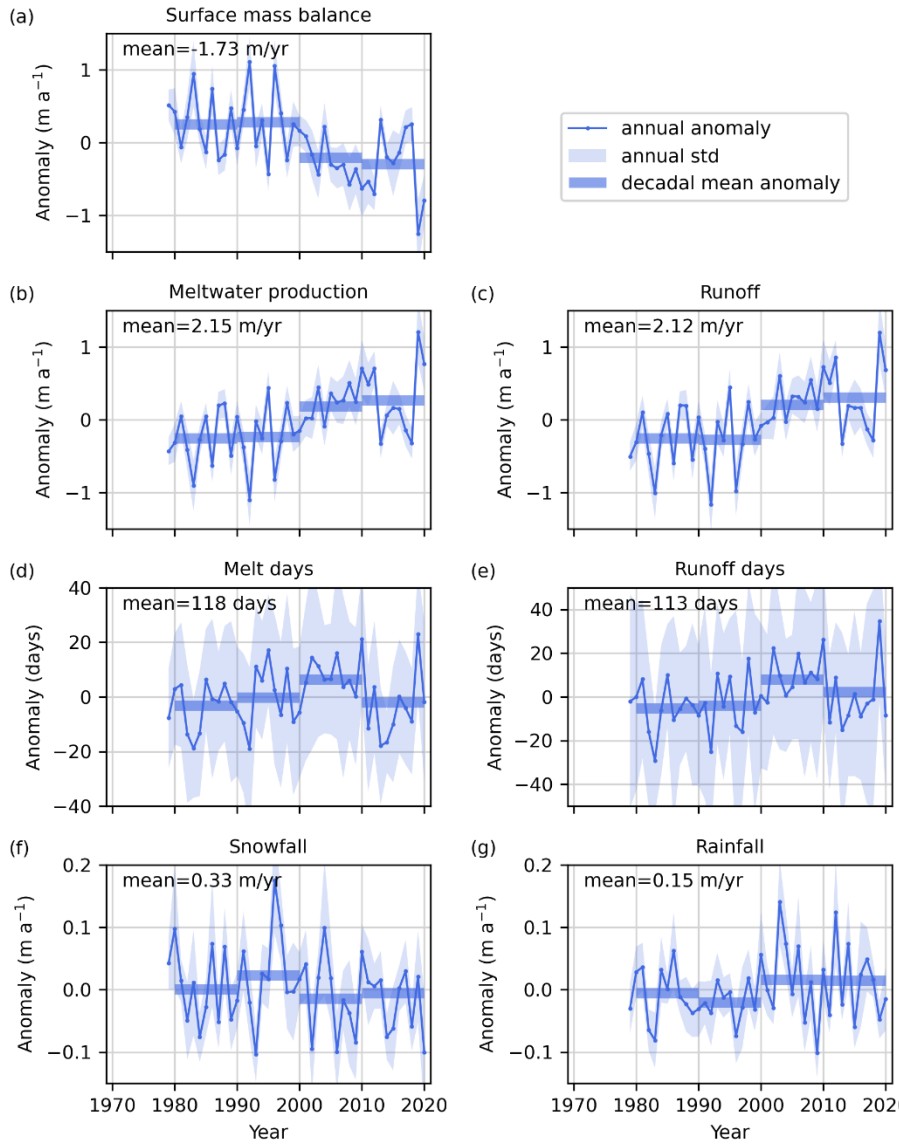

**Figure 6: Annual and decadal mean anomaly of (a) surface mass balance, (b) meltwater production, (c) number of melt days, (d) runoff, (e) number of runoff days, (f) snowfall, and (g) rainfall, from the MAR climate model, averaged over all glacier fronts in the study area. The mean for each anomaly is reported in each panel.**

Figure 7a shows Disko Bay warmed between 0.33°C (ICES) and 0.56°C (ECCO) between the 1990s and 2000s and cooled in 2016–2017 (the final years of the two subsurface temperature datasets). North of Disko Bay (Fig. 7b-h), however, the subsurface ocean temperatures followed a continuously increasing trend at every location we sampled. The temperature also warmed substantially (0.46–0.49 °C) between the 1990s and 2000s at Uummannaq Fjord, Wolstenholme Bay, and Inglefield Fjord (Fig. 7b, g, h), with a lesser magnitude of warming (0.06–0.11 °C) between the 2000s and the 2010s. In Upernavik Icefjord and throughout Melville Bay (Fig. 7c-f) temperatures oscillated around a steady decadal warming trend.





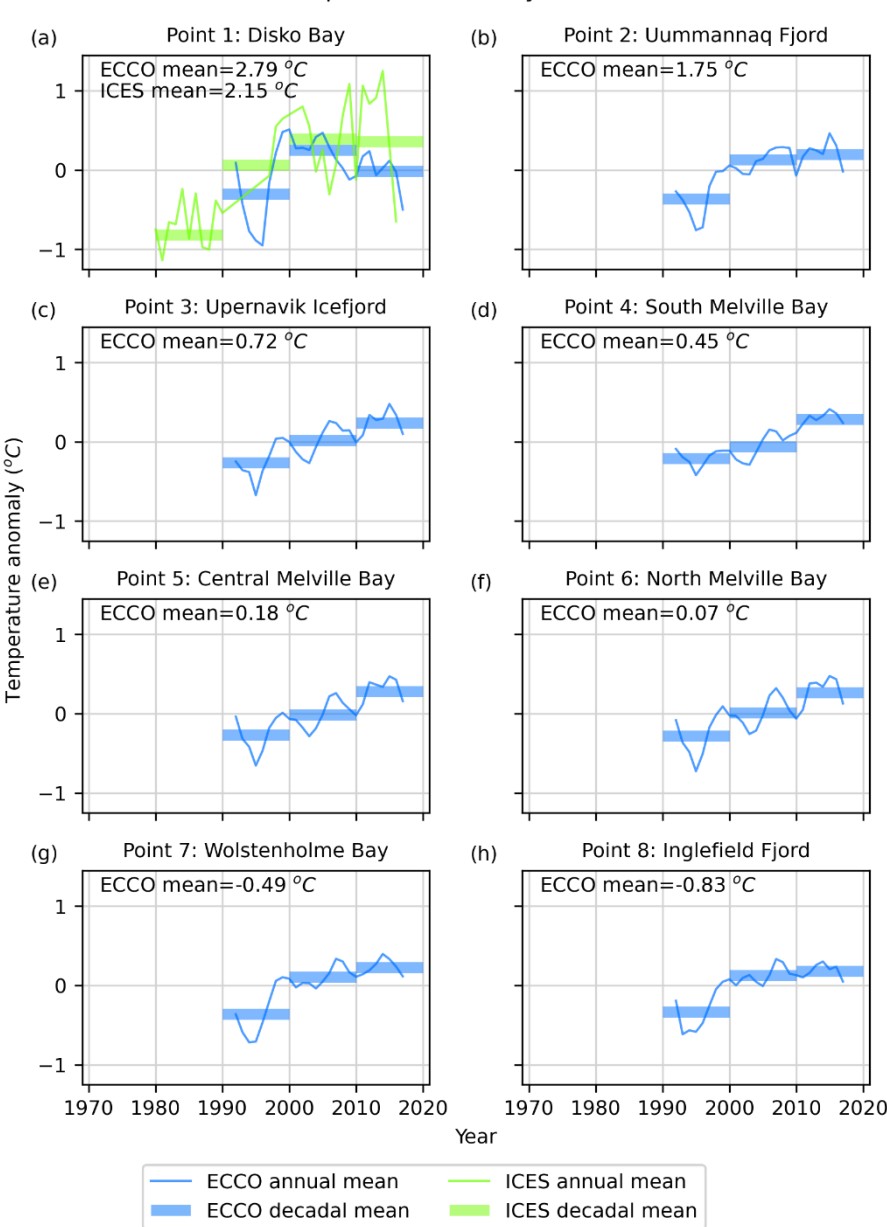

**Figure 7: Annual and decadal mean ocean temperature anomaly at 250 m depth from ECCO (blue) and ICES (purple). Each panel corresponds with an ocean point in Figure 1. The mean for each anomaly is reported in each panel.**

Figure 8 shows that sea-surface temperatures (SST) were relatively steady in the 1970s and 1980s. At some locations the SSTs dipped slightly in the 1990s, followed by a sharp rise (0.57–0.87 °C from ECCO; 0.25–0.81 °C from Hadley-OI) in SSTs in the 2000s at all sites. Surface temperatures in the 2010s were slightly warmer or cooler than in the 2000s, depending on the location and data source, but remained consistently above the pre-2000s temperatures.





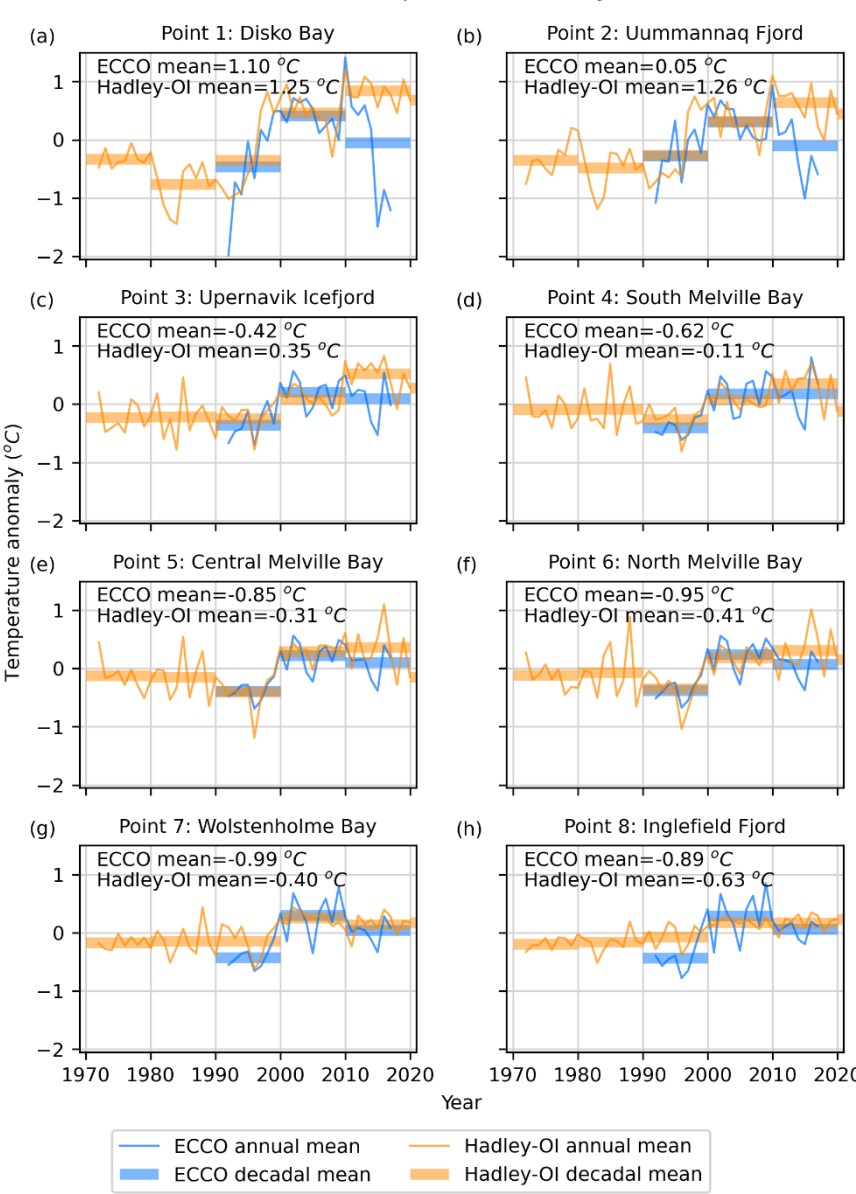

**Figure 8: Annual and decadal mean sea-surface temperature anomaly from ECCO (blue) and Hadley-OI (orange). Each panel corresponds with an ocean point in Figure 1. The mean for each anomaly is reported in each panel.**

Figure 9 shows that the annual duration of the sea-ice season (defined as when sea-ice concentration exceeds 15%) has
260    shortened since the 1970s. In particular, from Upernavik Icefjord north to Inglefield Fjord (Fig. 9c-h), while the decadal
mean sea-ice season length was relatively steady from the 1970s through the 1990s, it decreased by one to two months
between the 1990s and the 2000s, and remained relatively steady at the new shorter duration in the 2000s and 2010s. This
pattern is borne out in the seasonal sea-ice concentration as well at many locations (Supplementary Figs. 1-4).

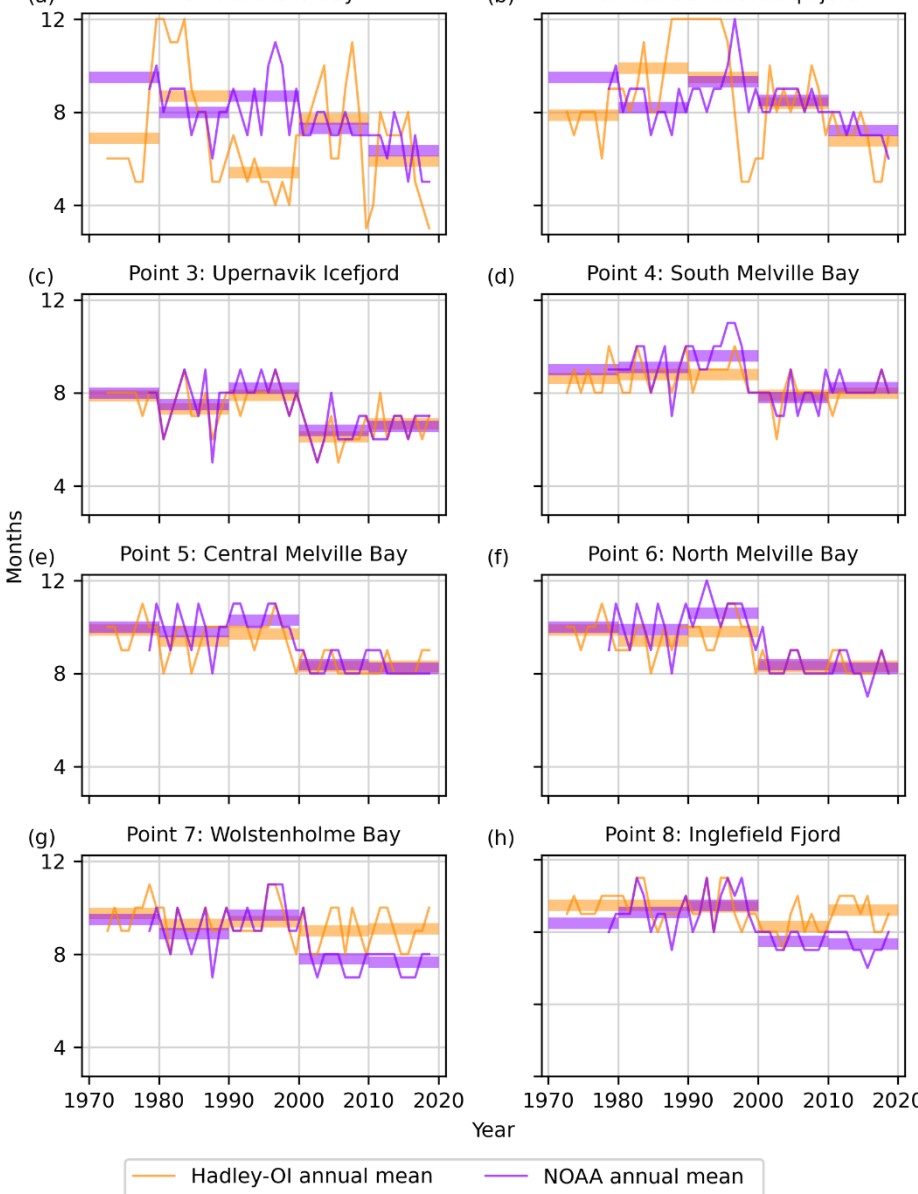

265 **Figure 9: Annual duration of sea-ice season (when sea-ice concentration is greater than 15%) from Hadley-OI (orange) and NOAA (green). Each panel corresponds with an ocean point in Figure 1.**

All of the climate variables that we surveyed showed long-term trends consistent with regional climate warming. Most of these variables, in most locations that we sampled, also showed a sharp change in their long-term trends between the 1990s and 2000s.



## 4 Discussion

Our measurements (Fig. 5) show a multi-decadal trend of regional glacier retreat with a step-change acceleration around 1996. The timing of this acceleration is consistent with previous studies of northwest and/or central-west Greenland that identified accelerated glacier retreat (Fahrner et al., 2021; Howat and Eddy, 2011; Carr et al., 2017; Wood et al., 2021; Catania et al., 2018) and ice speedup and discharge (King et al., 2020; Joughin et al., 2018b) beginning in the mid- to late 1990s. Nearly all glaciers in our study region retreated between the onset of this acceleration and the end of our observations in 2021, in line with other observations of sustained retreat in the 21st century (Fahrner et al., 2021; Bunce et al., 2018; Howat and Eddy, 2011; Murray et al., 2015). Howat and Eddy (2011) found that between 2000 and 2010, nearly 100% of glaciers in northwest Greenland retreated; although we considered a larger set of glaciers, we found that 85% of our glaciers retreated and 15% were stable over the same time period. Although 22% of glacier termini positions stabilized or remained stable between 2010 and 2020, 74% continued to retreat. Overall, between 2000 and 2020, 86% of glaciers retreated and 14% maintained stable terminus positions.

We observed a brief period of regional advance in 2017 and 2018, which was negated by regional retreat in 2019 (Fig. 5a,b). Khazendar *et al.* (2019) observed readvance of Jakobshavn Isbræ in 2017 and 2018 and attributed this behavior to regional ocean cooling at depth; however, while Jakobshavn Isbræ remained in a relatively advanced position in 2019 and 2020 and retreated in 2021 (Fig. 4a,b), the region as a whole began to retreat in 2019. Our observations indicate that glacier advance in 2017 and 2018 is more extensive than Jakobshavn, and in fact extends up the northwest coast. This regional behavior is coincident with a sustained positive surface mass balance anomaly in 2017 and 2018 with values near the 20th Century mean anomaly. This positive anomaly was followed by the strong negative anomalies in the record in 2019 and 2020 (Fig. 5a). However, we lack sufficient ocean temperature or sea ice data during this period to assess a relationship between those factors and regional glacier behavior. While not conclusive, the data do suggest SMB anomalies may have some effect on terminus position.

Several mechanisms have been proposed as drivers of outlet glacier retreat, including terminus ablation and undercutting, mélange rigidity, and enhanced hydrofracture, all of which can affect calving (Straneo et al., 2013). All of these mechanisms are tied to climatic warming in some sense, whether due to rising air or ocean temperatures, and associated changes in meltwater production, runoff, and sea-ice concentration. Our observed step-change acceleration in glacier retreat was approximately coincident with sharp increases in meltwater production (22%; Fig. 6b), runoff (26%; Fig. 6c), sea-surface temperature (0.25–0.87 °C; Fig. 8), and, in some regions, ocean subsurface temperature (0.46–0.49 °C; Fig. 7), and a sharp decrease in the duration of the sea-ice season (1–2 months; Fig. 9). Thus, any or all processes related to these anomalies could have contributed to the terminus retreat rates.

Ocean subsurface and surface warming in Baffin Bay and adjacent fjords have been cited as contributors to regional glacier retreat (Wood et al., 2021; Rignot et al., 2012; Slater et al., 2019) as well as for individual glaciers (Rignot et al., 2010; Holland et al., 2008; Motyka et al., 2011). Warmer ocean water increases melt at the calving face and, in conjunction with



subglacial discharge plumes, subsurface warming undercuts the terminus (Motyka et al., 2013; Slater et al., 2015, 2017), which could enhance calving (Luckman et al., 2015; Rignot et al., 2015; Morlighem et al., 2019; How et al., 2019). Wood *et al.* (2021) found that ocean thermal forcing switched from a stable period to one of rapid warming between 1997 and 1998, consistent with the timing of the rapid acceleration in glacier retreat we observe. Based on the same ocean dataset, we found that ocean subsurface temperatures increased an average of 0.37 °C coincident with our observed acceleration in glacier retreat (Fig. 7). The increase in ocean subsurface temperatures at that time appears larger in four subregions (Disko Bay, Uummannaq Fjord, Wolstenholme Bay/Thule, and Inglefield Fjord). Wood *et al.* (2021) identified deep glaciers sitting in warmer water as those that are retreating the most. Terminus response, however, tends to scale non-linearly with depth and deeper termini should be more responsive to any change that induces retreat (Schoof, 2007). As evidence of this effect, we note the strong response to seasonal perturbations on three of Greenland's deepest outlet glaciers (Jakobshavn, Kangerlussuaq, and Helheim), which is large compared to the degree of melting at the terminus (Kehrl et al., 2017; Joughin et al., 2020). Thus, irrespective of the forcing that caused the initial perturbation, a greater response would be expected for the deeper glaciers (>100 m depth), whether the forcing is due to ocean melting or some other process.

Similar to ocean subsurface temperatures, sea-surface temperatures averaged over all of our subregions also increased (0.58 °C) in concert with accelerated glacier retreat (Fig. 8). Fahrner (2021) found a significant relationship between sea-surface temperatures and terminus change in northwest Greenland, whereas Murray *et al.* (2015) found no relationship between retreat and sea-surface temperature in this region. Elsewhere in Greenland, warming sea-surface temperatures have been linked with rapid terminus retreat (Howat et al., 2008). Although warmer sea-surface temperatures are unlikely to contribute to submarine undercutting of the terminus, they may inhibit sea-ice formation and reduce mélange rigidity (Bevan et al., 2019).

Models indicate that the absence of a rigid mélange may be more important than ocean-driven melting of the terminus in enhancing glacier retreat (Todd and Christoffersen, 2014). Observations show a correlation between terminus change and sea ice and mélange conditions (Carr et al., 2017; Moon et al., 2015; Sohn et al., 1998), and that reduced sea ice and mélange formation could have triggered retreat at several Greenland glaciers (Howat et al., 2010; Amundson et al., 2010; Joughin et al., 2008a; Sohn et al., 1998). The presence of a rigid mélange – icebergs bound by a sea-ice matrix – can exert sufficient force to inhibit calving, facilitating seasonal glacier advance (Amundson et al., 2010; Cook et al., 2021; Cassotto et al., 2015; Robel, 2017; Reeh et al., 2001). We observed a sharp reduction in both the duration of the sea-ice season (Fig. 9) and of sea-ice concentrations (Supp. Figs. 1-4), coincident with an acceleration in regional glacier retreat in 1996; this correspondence indicates that a reduction in the presence and strength of mélange in glacier fjords may have occurred that could have increased calving leading to the accelerated retreat. Moreover, we observed a coincident increase of 26% in runoff and 11% in the number of runoff days (Fig. 6c,e), which seasonally may have contributed to more mobile mélange near the terminus. The observed increase in sea-surface and ocean subsurface temperatures could also inhibit sea-ice formation and strength; at Jakobshavn, periods of reduced mélange rigidity coincide with periods of warmer water at depth (Joughin et al., 2020). Thus,



the combination of reduced sea-ice duration and concentration, increased runoff, and increased sea surface and subsurface temperature together suggest a reduction of mélange presence and rigidity that might have increased calving and retreat.

We investigated the potential effect of a shortened sea-ice season on terminus positions, using observations of seasonal terminus range and mean velocity for sixteen glaciers in northwest Greenland – a subset of our study region – from Moon *et al.* (2015) as representative values. From their values, we estimated summer and winter calving rates, and used those values to estimate the mean calving rate in a state where the summer is two months longer, based on our observed two-month shortening of the sea-ice season (Appendix A). From these data, we estimate that increasing the summer duration by that much could increase the mean terminus retreat by 100 m $a^{-1}$, which would explain ~60% of the retreat observed by Moon *et al.* (2015). Although we assumed that the summer season was initially four months long, this result is independent of the initial summer duration. Our estimates indicate that changes in the duration of the sea-ice season could alone account for much of the observed retreat.

In addition to reducing mélange rigidity, increased meltwater production and runoff could also increase hydrofracture-induced calving. Crevasses filled with surface melt penetrate deeper than dry crevasses (Weertman, 1973; van der Veen, 1998), and if a crevasse near the terminus penetrates the full thickness of the ice it can cause calving (Nick et al., 2010; Sohn et al., 1998). We observed a sustained 26% increase in runoff (Fig. 6c) coincident with accelerated glacier retreat in 1996, which could contribute to increased filling of surface crevasses and subsequent hydrofracture, which would facilitate greater calving and retreat. Although runoff remained high in the 2010s, the duration of the runoff season decreased by 6 days (Fig. 6e) while sustained glacier retreat continued. These results suggest that to the extent that hydrofracture may have contributed to retreat, it is through increased runoff volume rather than seasonal duration.

## 5 Conclusions

We have built a comprehensive record of annual terminus positions for 87 marine-terminating outlet glaciers in northwest and central-west Greenland from 1972 through 2021. The majority of these glaciers retreated and lost area over the observation period, with retreat accelerating after 1996. We observed a brief regional readvance in 2017 and 2018, which was offset by losses in 2019. Ice-sheet climate data indicate that surface mass balance, meltwater production, and runoff increased substantially between the 1990s and 2000s, coincident with accelerated glacier retreat. Similarly, in most regions sea-surface temperature increased and sea-ice season duration decreased between the 1990s and 2000s, and in some regions ocean subsurface temperature also increased at this time.

Our results are consistent with other results indicating several potential mechanisms driving regional glacier retreat, and we cannot rule in or out any of them definitively. Furthermore, most of the climate factors that we looked at can influence more than one mechanism, and it is likely that some combination of these processes is at play. Thus, some combination of frontal melt, mélange presence and rigidity, and hydrofracture-induced calving is likely responsible for the widespread observed





retreat across northwest and central-west Greenland. Further research is needed to improve our understanding of the dominant processes contributing to terminus retreat and the resulting increases in discharge.

## Appendix A

We used data from Table 1 in Moon *et al.* (2015) to estimate seasonal calving rates in northwest Greenland and to assess the effect of increase the duration of the summer season. First, we determine the mean calving rate (Amundson and Truffer, 2010):

$$\frac{dX}{dt} = u_t - u_c - \dot{m}$$

where $X$ is the terminus position, $t$ is time, $u_t$ is the terminus velocity, $u_c$ is the calving rate, and $\dot{m}$ is the melt rate. Moon *et*

*al.* (2015) found a mean terminus velocity of 1647 m a⁻¹, and assuming steady-state conditions (no terminus change) and no melt, this gives us a calving rate of 1647 m a⁻¹.

Next, to estimate summer and winter calving rates, we assume that the calving rate takes the shape of a square wave. The amplitude of the square wave is equal to the average annual range in terminus positions, 610 m (Moon et al., 2015), and the mean value over one year is equal to our calving rate, 1647 m a⁻¹. Assuming an initial summer duration of 4 months, we

solve for the summer and winter calving rates:

$$\frac{4}{12} u_{c,summer} + \frac{8}{12} u_{c,winter} = 1647 \; m \; a^{-1}$$

$$u_{c,winter} = u_{c,summer} - 610$$

This yields a summer calving rate of 2054 m a⁻¹ and a winter calving rate of 1444 m a⁻¹.

We use these seasonal calving rates to estimate a new annual mean calving rate when the duration of the summer season has

increased by two months (or, effectively, the sea-ice season has decreased by two months).

$$\frac{6}{12} (2054 \; m \; a^{-1}) + \frac{6}{12} (1444 \; m \; a^{-1}) = 1749 \; m \; a^{-1}$$

This calving rate is 102 m a⁻¹ greater than the initial calving rate, indicating that increasing the duration of the summer, when calving rates are higher, by two months results in a 102 m a⁻¹ increase in calving. The mean terminus velocity value already accounts for an observed acceleration, so this increased calving rate is equivalent to an increase in terminus retreat. Moon *et*

*al.* (2015) found a terminus retreat of 920 m over ~5.5 years (~167 m a⁻¹), so our 102 m a⁻¹ increase in calving due to increased summer duration can account for 61% of their observed retreat. Other processes that extend the length of the calving season (e.g., a longer subglacial discharge season) could also explain the observed increase in the duration of the summer calving season.

We repeated these calculations with different assumptions for the initial summer duration and found that the results are

independent of this assumption.



**Code Availability**

Data analysis and visualization code are available on GitHub (https://github.com/tarynblack/northwest_decadal_2021).

**Data Availability**

The terminus positions will be made available on NSIDC as part of the MEaSUREs Annual Greenland Outlet Glacier
Terminus Positions from SAR Mosaics dataset (https://nsidc.org/data/NSIDC-0642). Bed topography is from BedMachine
Greenland V3 (https://nsidc.org/data/idbmg4). Ice-sheet velocity is from MEaSUREs Multi-year Greenland Ice Sheet
Velocity Mosaic, Version 1 (https://nsidc.org/data/NSIDC-0670/versions/1). Landsat images were identified with GloVis
(http://glovis.usgs.gov) and downloaded from Google Cloud Platform
(https://console.cloud.google.com/storage/browser/gcp-public-data-landsat). Sentinel-1 images are from MEaSUREs
Greenland Image Mosaics from Sentinel-1A and -1B, Version 3 (https://nsidc.org/data/nsidc-0723). Ocean temperatures are
from ECCO Version 5 (https://ecco-group.org), the ICES Dataset on Ocean Hydrography
(https://ocean.ices.dk/hydchem/hydchem.aspx), and the Merged Hadley-OI sea surface temperature and sea ice concentration
data set (https://doi.org/10.5065/r33v-sv91). Sea-ice concentrations are from the Merged Hadley-OI data set and the
NOAA/NSIDC Climate Data Record of Passive Microwave Sea Ice Concentration, Version 3
(https://nsidc.org/data/g02202). Ice sheet climate data are from MAR v3.11 output over Greenland
(ftp://ftp.climato.be/fettweis/MARv3.11/Greenland/ERA_1979-2020-6km/).

**Author Contribution**

T.B. and I.J. conceptualized the project. T.B. carried out the terminus data collection, analysis, and visualization. I.J.
prepared the SAR data products. T.B. prepared the manuscript, with contributions from I.J.

**Competing Interests**

The authors declare that they have no conflict of interest.

**Acknowledgements**

T.B. and I.J. were supported by the NASA MEaSUREs program (80NSSC18M0078). T.B. was also supported by an NSF
Graduate Research Fellowship early in the project. Twila Moon digitized the original six-year dataset of terminus positions.



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
