# Peer review of "Multi-decadal retreat of marine-terminating outlet glaciers in northwest and central-west Greenland"

_The Cryosphere, 2021_

## Author Response (AR1)

**Response to Reviewer #1 (Michael Wood) on "Multi-decadal retreat of marine-terminating outlet glaciers in northwest and central-west Greenland"**

Comment received: 19 July 2021

Key:

*Reviewer comment (blue italics)*
Response (black)
* * *
Thank you for these detailed comments, which have greatly improved the paper.

*In this manuscript, Black and Joughin present a comprehensive record of glacier retreat in northwest and central-west Greenland, encompassing more than 80 glaciers across almost 5 decades. The results are compared with a variety of observations and models on the ice sheet and in the regional ocean to draw conclusions about the regional drivers of the observed glacier retreat. This study is timely and of scientific interest because terminus retreat in this region is linked with dynamics mass loss from the Greenland Ice Sheet – a substantial contributor to recent sea level rise.*

*The quantification of terminus retreat is a particularly robust component of this study. The record extends previous data of ice front position both forward and backward in time, and it supports previous findings about the magnitude and timing of overall glacier retreat in this region. Another particularly compelling component of this study is the authors' approach to investigate a wide variety of model and observational data for regional changes that have the potential to influence glacier retreat.*

Summary comments, no action taken.

*One major concern I had in reading this manuscript is that the main conclusion of this paper – "that a variety of processes … contribute to, but do not conclusively dominate, the observed regional retreat" – is not sufficiently supported by the analysis of the data presented. In particular, two main points remain unclear: 1) the extent to which the sampled data and model results on the continental shelf reflect conditions within the fjords and near the glacier termini, and 2) a quantitative link between the documented regional parameters and glacier retreat.*

These concerns are described in more detail below and we address them there.

*For oceanographic data and model results, there are 8 sampling locations on the continental shelf, chosen to be representative of different "clusters" of glaciers along the coastline. The authors point toward a lack of reliable data in narrow fjords as justification for using offshore data as a proxy for fjord conditions; however, for ocean observations in particular, NASA's Ocean's Melting Greenland mission has been conducting conductivity-temperature-depth surveys of the fjords and continental shelf since 2015. This publicly-available dataset provides a means to quantify the extent to which oceanographic conditions on the continental shelf relate to conditions*

*within the fjords, as well as assessing the ability of the 1/3 degree ECCO model to simulate ocean temperature on the continental shelf.*

Because the paper focuses on annual trends over several decades, we chose to use longer-term oceanographic datasets that could provide annual averages, rather than the OMG data, which do not go back as far in time and provide only seasonal snapshots. With regards to the continental shelf vs. fjord conditions, in revising the paper we made several adjustments to our subsurface ocean temperatures. Emulating your Wood et al. (2021), we 1) calculated the depth-averaged ocean temperature over the bottom 60% of the water column, instead of using the 250m temperature, in order to better capture the ocean heat at depth; 2) we then calibrated those data to match the temperatures in comparable regions provided in the supplementary data of Wood et al. (2021), which performed a more detailed analysis of offshore and fjord ocean temperatures.

*In a similar sense, it is unclear the extent to which changes in sea ice concentration and sea surface temperature outside of the fjords reflect changes in rigidity of ice mélange near the glaciers. Previous studies have used optical data and/or feature tracking of icebergs within fjords to get a sense of mélange rigidity, but it is unclear how these results compare with sea ice concentration and surface temperature outside of the fjords. Overall, an explicit quantitative link between conditions on the shelf and those in the fjords where they interact with glacier ice fronts would make the analysis in this study more robust.*

We are using the sea ice concentration and sea surface temperature outside the fjords as a proxy – albeit, an imperfect one – for water in the fjords, which will in turn influence changes in mélange. The link between sea ice conditions and mélange rigidity has been previously shown (*e.g.*, Joughin et al., 2008a; Amundson et al., 2010). We will also be looking at more detailed datasets of mélange rigidity inside the fjords as a subject of future work.

*For the impact on glacier retreat, this study presents a suite of possible conditions which could theoretically induce changes in the glacier front positions through variations in the rates of iceberg calving and submarine melt. However, the analysis lacks a quantitative comparison with glacier retreat, either statistical or mechanical. Statistically, for example, how well do variations in glacier front positions correlate with any of the given parameters, either on a glacier-by-glacier, a glacier "cluster", or a regional level? While a visual comparison can be drawn by flipping through Figs 5-9, an explicit analysis, such as a multiple regression, would help more firmly establish this link and reveal which processes might be more prominent. From a mechanistic standpoint, how would the impact of the various regional changes compare quantitatively? For example, given the changes in regional oceanographic and atmospheric conditions, how would the magnitude of retreat from ocean melt (e.g. as a result of warmer ocean conditions and higher subglacial discharge) compare to the magnitude of retreat from enhanced calving as related to a decline in ice mélange rigidity (e.g. as a result of warmer surface temperature and lower sea ice concentration)? A quantitative comparison of all the potential processes listed would help support the conclusion that any particular processes does not conclusively dominate retreat.*

We performed a suite of multiple linear regressions to test the relationships between our various oceanographic and ice-sheet variables and terminus retreat and have incorporated the results into the discussion section, including a table of the resulting sensitivities to different climate variables. The regression results indicated that terminus retreat is most sensitive to runoff, and moderately sensitive to ocean temperatures, but only weakly corresponds to sea ice. We have revised the discussion accordingly. We feel that this has made the conclusions much more robust and we appreciate the suggestion.

*Overall, this is an important study seeking to identify regional drivers of glacier retreat in Greenland, helping to shape our understanding of recent and future sea level rise contributions from the Greenland ice sheet. However, I would recommend a revision of the paper to provide a more robust, quantitative link between the regional conditions investigated in this study and the history of glacier retreat.*

We believe that the revisions made in response to this review have addressed these quantitative concerns and greatly improved the paper.

**Response to Reviewer #2 (anonymous) on "Multi-decadal retreat of marine-terminating outlet glaciers in northwest and central-west Greenland"**

Comment received: 5 August 2021

Key:
*Reviewer comment (blue italics)*
Response (black)

Thank you for these detailed comments, which have greatly improved the paper.

*Summary:*

*The authors analyzed a variety of environmental datasets in conjunction with annual records of terminus position changes from 1972-2021 for 87 marine-terminating glaciers in western Greenland. They find that there was a step change in terminus positions in ~1996 and that nearly all glaciers have retreated since the turn of the century. Changes in retreat coincide with a number of environmental parameters so the authors cannot conclusively point to one trigger for the observed retreat. There are a few places where the presentation of the methods and discussion can be modified to improve clarity, as outlined below, but there are no major methodological flaws or issues with interpretation. Overall the paper is easy to read and presents an interesting analysis that bridges the gap between the shorter-term but highly detailed analyses and long-term but broad analyses commonly found in the literature.*

Summary comment, no action taken.

*Major Points:*

1. *There are some parts of the methods that need a bit more detail.*

    7. *The first dataset that would benefit from more detail is the terminus position dataset. For example, when Landsat 7 SLC-off data are available, are those preferred to the Landsat 5 images despite the image gaps simply because they are higher resolution images?*

        During the time periods for which we used Landsat-7 images, there were generally no Landsat-5 images available. Although Landsat-5 was in orbit at the time, it didn't continuously collect images outside of the U.S. and may not have been prioritized over Greenland while Landsat-7 was also in orbit. Added, "We continued to use Landsat-7 images after the instrument's scan-line corrector failure in 2003 as Landsat-5 images were typically unavailable over our study area during this period, although we only kept images which retained a sufficient amount of data to map the terminus. In those images, we digitized across the scan-line corrector gaps when they crossed a glacier terminus; in all, the scan-line corrector gaps affected 348 terminus traces (9.7% of the dataset)."

        *Also, for the optical images, what bands were used? I assume panchromatic for Landsat 7 and 8 but that band does not exist before Landsat 7.*

Added, "We digitized termini using the panchromatic band for Landsat-7 and Landsat-8, and using a single band that provided high image contrast (typically band 2) for Landsat 1-5."

*Figure 2 should also be referenced in this section since it conveys important information regarding the seasonality of the satellite images.*

Done.

8. *For the environmental data, I recommend you move much of the information in the last paragraph of section 2.4 to the end of the first paragraph or immediately after it. It is helpful for the reader to know what metrics you extracted from each dataset before they read through the details for each dataset.*

The final paragraph has been moved up to the end of the first paragraph as suggested.

*It is also helpful to know more about the location from which the ice sheet variables were extracted. Is "near the front" a fixed location within a certain distance from the most retreated terminus position? Does the position move with the terminus?*

Data were extracted from a fixed location near the front; we added this to the text.

*Did you consider extracting these data from the full catchment for each glacier?*

No, because the most relevant processes for this work are occurring at the terminus; changes farther upstream take some time to propagate to the terminus, so we focused on the regions near the terminus.

*For the ocean data, why did you pick a depth of 250 m? Does the depth selected influence your interpretation?*

We had chosen a depth consistent with that used in other papers, which should allow water to enter fjords even with relatively shallow sills (*e.g.* in Disko Bay). However, in response to this comment we revisited the temperature depth profiles, and decided to instead use the depth-averaged temperature over the bottom 60% of the water column to capture the mean temperatures likely reaching glacier fronts (while avoiding surface waters, which we examined separately).

9. *For each climate variable, more justification for its use should be presented earlier in the manuscript. There is some discussion about the importance of each variable, but justification should be presented in the methods so that the reader understands why these data are used in the comparison.*

At the beginning of section 2.4, we added, "Earlier work has shown that climate-related processes including terminus ablation and undercutting (driven by ocean

warming), mélange rigidity (driven by changes in ocean temperature, sea ice concentration, and/or runoff), and enhanced hydrofracture (driven by changes in runoff and surface mass balance) may affect terminus position. Hence, we acquired…"

2. *In the example in Figure 3, it looks like the glacier margin may expose land along its southern margin or at least a region of very stable ice. The inclusion of this essentially stagnant ice margin may considerably influence your terminus change rate calculations. Although you acknowledge that the boxes were drawn somewhat arbitrarily, it may be helpful to impose a velocity threshold to identify regions of stagnant ice that will artificially lower retreat rates if included in some terminus boxes and not others.*

The change measurements specifically calculate area change (ice area gained/lost) within the box, and any zones that do not change are effectively ignored. That is, the box could be drawn to any arbitrary shape, including stagnant ice, and that would have no effect on the end result (total ice area gained/lost). In this particular case, the now-stagnant zone used to be part of an active margin, so we needed to place the box along that zone in order to bound all of the terminus traces. We added a brief note in the text to clarify this point.

3. *For the break-point analysis, did you include data for all years? Did you include the linearly interpolated terminus position during observation gaps? In looking at the seasonal data coverage in Figure 2, it looks like 1993 and 1995 were years when most terminus observations are from summer imagery (although it is really difficult to distinguish the summer and autumn colors). Have you considered that the inclusion of these summer observations in relatively rapid succession may have an influence on the break point analysis?*

Yes, the break-point analysis includes data from all years, including the linearly interpolated terminus positions. Running the analysis with only the observed (non-interpolated) areas does not substantively change the results. Regarding the summer observations, we considered this during the original analysis, but took another look in response to this comment and there appears to be no discernable effect.

(An aside on the color scale – it was specifically chosen to be cyclical, since late summer becomes early autumn, so the colors of those periods will be similar)

4. *I thought the results were presented in a clear and logical order but the discussion is a little disorganized. Like the section describing the climate data, it would be helpful to have an overview of your data synthesis at the beginning of the discussion instead of buried in lines 292-299. Then you can dedicate a paragraph to observed relationships between retreat and each variable and the reader will be able to follow why those variables should be related.*

We moved the noted paragraph to the beginning of the discussion and added subheadings throughout the discussion to clarify the organization.

*Minor Comments:*

- *line 24: Rearrange this sentence slightly from "in northwest and central-west Greenland over half of the mass loss is currently due to ice discharge" to "over half of the mass loss in northwest and central-west Greenland is currently due to ice discharge"*

Done.

- *lines 36-38: The phrase "Because of the interaction of terminus position with glacier geometry" is somewhat awkward. I recommend dropping that portion of the sentence and slightly rephrasing the remainder of the sentence so it can be merged with the previous sentence.*

Rewrote and merged with previous sentence as follows: "In turn, responses to these forcings are modulated by geometric factors associated with individual glaciers such as bed topography and fjord width (Carr et al., 2015; Schild and Hamilton, 2013; Catania et al., 2018; Felikson et al., 2021); for many glaciers, these modulating factors necessitate detailed records of terminus position changes in order to identify the importance of different forcing mechanisms on decadal-scale outlet glacier changes across a large area."

- *lines 83-84: Are these size thresholds based on previous analyses? How were they determined?*

Rewrote as "To focus our analysis on glaciers that produce substantial discharge, we limited our analysis to marine-terminating outlet glaciers that are at least ~1.5 km wide at the terminus and are flowing at least ~1000 m a$^{-1}$." This is consistent with other studies.

- *In section 2.2, you mention that errors are from the imagery and digitization but then present the sources of error in the opposite order.*

The error sources are presented in the order in which they are originally mentioned. Changed "Manual tracing introduces errors…" to "Manual digitization also introduces errors…" to clear confusion.

- *Why is Figure 2 missing data for all years for 2-3 glaciers (77?-78)?*

The glaciers ID numbers come from a pre-existing dataset, and we excluded glaciers 76-78 because they are disconnected from the main ice sheet (i.e. they are not ice-sheet outlet glaciers). Added to the caption, "Glaciers #76-78 are not shown because they are not ice-sheet outlet glaciers and therefore are not included in this study."

- *line 147: Change "singular important" with the "single-most important"*

Done.

- *Figure 5: Change the initial description from "Timing of change" to something more descriptive.*

Changed to "Break-point analysis to identify the onset of increased terminus retreat rates"

- *I really like the MAR panels in Figure 6.*

Thanks!

- *lines 278-281: Does the same population of glaciers decrease from 2000-2010 and 2010-2020?*

Good clarification, we ran the numbers and rewrote those lines as follows: "Between 2010 and 2020, 74% of glaciers retreated and 22% were stable. Overall, between 2000 and 2020, 86% of glaciers retreated and 14% maintained stable terminus positions; 69% of glaciers retreated in both decades and 9% were stable in both decades."

- *line 296 and elsewhere: I find the expression "step-change acceleration in glacier retreat" to be misleading. My interpretation of your data is that there was a sudden onset of terminus retreat for most glaciers in 1996. I would describe this as a "step-change in terminus retreat rate" or "acceleration in terminus retreat rate" if the termini were generally retreating before 1996 but at a slower rate.*

We have adopted "step-change in terminus retreat rate" throughout the text.

- *line 327: You don't define ice mélange until this line but refer to it far earlier in the text.*

Good catch, moved the definition up to the first mention of mélange in the introduction: "…changes in the characteristics of sea ice and mélange – icebergs bound by a sea-ice matrix…"

- *lines 338-346 and appendix: The description of the effect of sea ice of terminus positions is really difficult to follow in just the main text. I'm left wondering why this analysis is mostly confined to the appendix and not woven into the rest of the manuscript.*

In response to another review, we performed a regression analysis on our climate variables and found that the effect of offshore sea ice on terminus positions was weaker than we previously expected. Consequently, we removed the appendix about the effect of sea ice on terminus positions.

---

## Author Response (AR2)

**Response to Editor Report on "Multi-decadal retreat of marine-terminating outlet glaciers in northwest and central-west Greenland"**

Report received: 24 January 2022

Key:
*Report comment (blue italics)*
Response (black)
* * *
*1. add the suggested discussion on what additional information would be needed to more conclusively reveal the primarily driver(s) of glacier retreat*

We added a sentence to the conclusion addressing additional information (mélange, oceanographic data) that would be helpful in clarifying the primary drivers of glacier retreat.

We apologize for not previously addressing the comments from the initial editor review, this was an oversight.

*2. change the color map of Fig.1b to be more scientifically accurate. Although I know it is a common format, rainbow maps should never be used. For the motivation of my comment and potential solutions, check: https://www.nature.com/articles/s41467-020-19160-7?s=09*

We have changed the color map to the matplotlib 'plasma' color map, which is perceptually uniform. We also note that in all other figures in this paper, color had been carefully chosen to be scientifically accurate.

*3. Fig5b. it is not fair to include the break-points even if no break-point is detected as it is a logical statistical artefact to have more fake break points in the middle of the time series. These not-significant breakpoints should be removed from the analysis.*

We added an F-test to determine whether the piecewise-linear fit (with breakpoints) is statistically better than a linear regression (without breakpoints) for each glacier, and discarded breakpoints that failed the F-test. We added a description of this process to the methods section, and updated Figure 5b and the breakpoints in Supplementary Table 3 accordingly.